# Nonlinear analysis of the occurrence of hurricanes in the Gulf of Mexico and the Caribbean Sea

Berenice Rojo-Garibaldi[1], David Alberto Salas-de-León[2], María Adela Monreal-Gómez[2], Norma Leticia Sánchez-Santillán[3], and David Salas-Monreal[4]

[1]Posgrado en Ciencias del Mar y Limnología, Universidad Nacional Autónoma de México, Av. Universidad 3000, Col. Copilco, Del. Coyoacán, Cd. Mx. 04510, México.

[2]Instituto de Ciencias del Mar y Limnología, Universidad Nacional Autónoma de México, Av. Universidad 3000, Col. Copilco, Del. Coyoacán, Cd. Mx. 04510, México.

[3]Departamento El Hombre y su Ambiente, Universidad Autónoma Metropolitana, Calz. del Hueso 1100, Del. Coyoacán, Villa Quietud, Cd.Mx. 04960, México.

[4]Instituto de Ciencias Marinas y Pesquerías, Universidad Veracruzana, Hidalgo No. 617, Col. Río Jamapa, C.P. 94290 Boca del Río, Veracruz, México.

Correspondence to D. A. Salas-de-León (dsalas@unam.mx)

**Abstract.** Hurricanes are complex systems that carry large amounts of energy. Their impact often produces natural disasters involving the loss of human lives and materials, such as infrastructure, valued in billions of US dollars. However, not everything about hurricanes is negative, as hurricanes are the main source of rainwater for the regions where they develop. This study shows a nonlinear analysis of the time series obtained from 1749 to 2012 of the occurrence of hurricanes in the Gulf of Mexico and the Caribbean Sea. The construction of the hurricane time series was carried out based on the hurricane database of the North Atlantic-basin Hurricane Database (HURDAT) and the published historical information. The hurricane time series provides a unique historical record on information about ocean-atmosphere interactions. The Lyapunov exponent indicated that the system presented chaotic dynamics, and the spectral analysis and nonlinear analyses of the time series of the hurricanes showed chaotic edge behavior. One possible explanation for this edge is the individual chaotic behavior of hurricanes, either by category or individually, regardless of their category and their behavior on a regular basis.

## Introduction


Hurricanes have been studied since ancient times, and their activity is related to disasters and loss
of life. In recent years, there has been considerable progress in predicting their trajectory and
intensity once tracking them has begun, as well as their number and intensity from one year to the
next. However, their long-term and very short-term prediction remains a challenge (Halsey and
Jensen, 2004), and the damage to both materials and lives remains considerable. Therefore, it is
important to make a greater effort in the study of hurricanes to reduce the damage they cause. The
periodic behavior of hurricanes and their relationships with other natural phenomena have usually
been performed with linear-type analyzes which have provided valuable information. However,
we decided to make a different contribution by carrying out a nonlinear analysis of a time series of
hurricanes that occurred in the Gulf of Mexico and the Caribbean Sea, since the dynamics of the
system is controlled by a set of variables of low dimensionality (Gratrix and Elgin, 2004;
Broomhead and King, 1986).
One of the core parts of this work was the elaborate time series that was built, especially for the
oldest part of the registry, in which it was possible to have a substantial and robust collection. This
gave our time series a number of data with which it was possible to perform this analysis;
otherwise, it would have been impossible to study this natural phenomenon with a nonlinear
analysis.
Different methods have been used in the analysis of non-linear, non-stationary and non-Gaussian
processes, including artificial neural networks (ASCE Task Committee, 2000, Maier and Dandy,
2000, Maier et al., 2010, Taormina et al. 2015). Chen et al. (2015) use a hybrid neural network
model to forecast the flow of the Altamaha River in Georgia; Gholami et al. (2015) simulate
groundwater levels using dendrochronology and an artificial neural network model for the
southern Caspian coast in Iran. On the other hand, theories of deterministic chaos and fractal
structure have already been applied to atmospheric boundary data (Tsonis and Elsner, 1988; Zeng
et al., 1992), e.g., to the pulse of severe rain time series (Sharifi et al., 1990; Zeng et al., 1992) and
to the tropical cyclone trajectory (Fraedrich and Leslie, 1989; Fraedrich et al., 1990). Natural
phenomena occur in nature within different contexts; however, they often exhibit common
characteristics, or they may be understood using similar concepts. Deterministic chaos and fractal
structure in dissipative dynamical systems are among the most important nonlinear paradigms
(Zeng et al., 1992). For a detailed analysis of deterministic chaos, the Lyapunov exponent is
utilized as a key point and several methods have been developed to calculate it. It is possible to
define different Lyapunov exponents for a dynamic system. The maximal Lyapunov exponent can
be determined without the explicit construction of a time-series model. A reliable characterization
requires that the independence of the embedded parameters and the exponential law for the growth
of distances can be explicitly tested (Rigney et al., 1993; Rosenstein et al., 1993). This exponent
provides a qualitative characterization of the dynamic behavior and the predictability measurement
(Atari et al., 2003). The algorithms usually employed to obtain the Lyapunov exponent are those
proposed by Wolf (1986), Eckmann and Ruelle (1992), Kantz (1994) and Rosenstein et al. (1993).
The methods of Wolf (1986) and Eckmann and Ruelle (1992) assume that the data source is
indeed a deterministic dynamic system and that irregular fluctuations in time-series data are due to
deterministic chaos. A blind application of this algorithm to an arbitrary set of data will always
produce numbers, i.e., these methods do not provide a strong test of whether the calculated
numbers can actually be interpreted as Lyapunov exponents of a deterministic system (Kantz et
al., 2013). The Rosenstein et al. (1993) method follows directly from the definition of the
Lyapunov maximal exponent and is accurate because it takes advantage of all available data. The
algorithm is fast, easy to implement, and robust to changes in the following quantities: embedded
dimensions, data-set size, delay reconstruction, and noise level. The Kantz (1994) algorithm is
similar to that of Rosenstein et al. (1993).
We constructed a database of occurrences of hurricanes in the Gulf of Mexico and the Caribbean
Sea to perform a nonlinear analysis of the time series, the results of which can help in the
construction of hurricane occurrence models, and this in turn will help to reinforce the measures of
prevention for this type of hydrometeorological phenomenon.

**2 Materials and methods**
**2.1 Data set description**
A detailed analysis of the historical reports, provided by the ships that were used, was carried out
in order to obtain the annual time series of the occurrence of hurricanes, from category one to five
on the Saffir-Simpson scale, in the study region from 1749-2012. The time series was composed
by the historical ship track of all vessels sailing close to registered hurricanes, the aerial
reconnaissance data for hurricanes since 1944 and the hurricanes reported by Fernández-Partagas
and Díaz (1995a, 1995b; 1996a; 1996b; 1996c; 1997; 1999). All this information in addition to the
database of the HURDAT re-analysis project (HURDAT is the official record of the United States
for tropical storms and hurricanes occurring in the Atlantic Ocean, Gulf of Mexico and Caribbean
Sea) was used in a comparative way in order to build our time series, which is so far the longest
time series of hurricanes for the Gulf of Mexico and the Caribbean Sea. This makes our series
ideal for performing a nonlinear analysis, which would be impossible with the records that are
available in other regions. All this information was used to build the hurricane time series (Fig. 1).

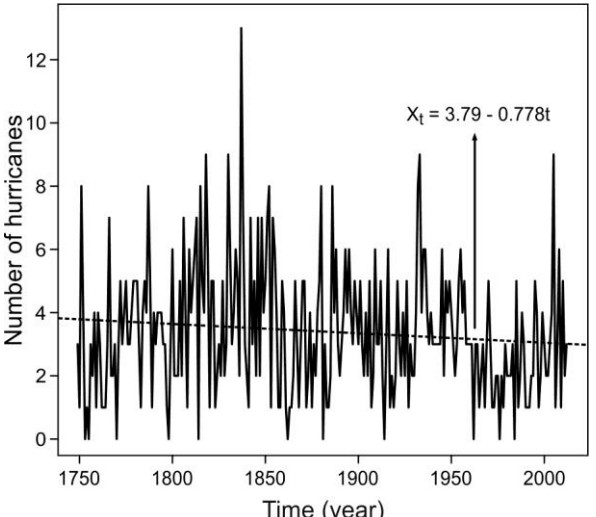


Figure 1. Hurricanes in the years 1749-2012. The line shows their linear trend (after Rojo-
Garibaldi et al., 2016).

Historical hurricanes were included only if they were reported in two or more databases and met
both of the following criteria: the reported hurricanes that touched land and those that remained in
the ocean; on the other hand, the followed hurricanes were studied considering their average
duration and their maximum time (9 and 19 days, respectively). This was done in order to avoid
counting more than one specific hurricane reported in different places within a short period time;
to do this, we followed the proposed method by Rojo-Garibaldi et al. (2016).

## 2.2 Data reduction and procedures

Before performing the nonlinear analysis of the time series, we removed the trend; thus, the series
was prepared according to what is required for this type of analysis. To know the properties of the
system requires more than estimating the dimensions of the attractor (Jensen et al., 1985); so, three
methods were applied in this study: 1) The Hurst exponent is a measure of the independence of the
time series as an element to distinguish a fractal series. It is basically a statistical method that
provides the number of occurrences of rare events and is usually called re-scaling (RS) rank
analysis (Gutiérrez, 2008); in addition, according to Miramontes and Rohani (1998), the Hurst
exponent provides another approximation that can be used to characterize the color of noise, and
therefore, it could be applied to any time series. The RS helps to find the Hurst exponent, which
provides the numerical value that makes it possible to determine the autocorrelation in a data
series. 2) The Lyapunov exponent is invariant under soft transformations, because it describes
long-term behavior, providing an objective characterization of the corresponding dynamics (Kantz
and Schreiber, 2004). The presence of chaos in dynamic systems can be solved by this exponent,
since it quantifies the exponential convergence or divergence of initially close trajectories in the
state space and estimates the amount of chaos in a system (Rosenstein et al., 1993; Haken, 1981;
Wolf, 1986). The Lyapunov exponent ($\lambda$) can take one of the following four values: $\lambda < 0$
corresponds to a stable fixed point, $\lambda = 0$ is for a stable limit cycle, $0 < \lambda < \infty$ indicates chaos and
$\lambda = \infty$ is a Brownian process, which agrees with the fact that the entropy of a stochastic process is
infinite (Kantz and Schreiber, 2004). 3) The iterated function analysis (IFS) is an easier and
simpler way to visualize the fine structure of the time series because it can reveal correlations in
the data and help to characterize its color, referring to color to the type of noise (Miramontes et al.,
2001). Together with the Lyapunov exponent, the phase diagrams, the False Close Neighbors
method, the Space-Time Separation plot, the Correlation Integral plot, and the Correlation
Dimension were taken into account, the latter two to identify whether the system attractor was a
fractal type. It is important to compute the Lyapunov exponent, so we used the algorithms
proposed by Kantz (1994) and Rosenstein et al. (1993) to calculate it.

## 3 Results and discussions

Figure 1 shows the evolution of the number of hurricanes from 1749 to 2012 and the linear trend. To have a qualitative idea of the behavior of the number of hurricanes that occurred in the Gulf of Mexico and the Caribbean Sea from 1749 to 2012, a phase diagram was performed using the "delay method" (Fig. 2). This was also used to elucidate the time lag for an optimal embedding in the dataset. The optimal time lag ($\tau$) obtained visually from Fig. 2 was equal to 9, since it was the time in which the curves of the system were better divided. We must not forget that this was only a visual inspection, and the delay time will be obtained quantitatively through other methods. In our case, the hurricane dynamics were not distinguished through the phase diagram; however, since any hurricane trajectory starts at a close point location on the attractor dataset that diverges exponentially, the phase diagram is a primary evidence of a chaotic motion according to Thompson and Stewart (1986).

The most robust method to identify chaos within the system is the Lyapunov exponent. Prior to obtaining the exponent, it was necessary to calculate the time lag and the embedding dimension, and for the latter, the window of Theiler was used. The time lag was obtained with three different methods: 1) the method of constructing delays, which is observed visually in Figure 2; 2) the method of mutual information, which yields a more reliable result since it takes into account nonlinear dynamic correlations; here, the delay time was obtained by taking the first minimum of the function; in this case $\tau = 9$; and 3) the autocorrelation function method, which is based solely on linear statistics (Fig. 3).

There are two ways to obtain the time lag from the autocorrelation function: 1) the first zero of the function, and 2) the moment in which the autocorrelation function decays as $1/e$ (Kantz and Schreiber, 2004). We used the criterion of the first zero because the Hurst exponent ($H = 0.032$) indicated that it was a short memory process; therefore, the criterion of the first zero is the optimal method in this type of case. By this method, the value that was obtained was $\tau = 10$. The value of this parameter is very important, since if it turns out to be very small, then each coordinate is almost the same and the reconstructed trajectories look like a line (the phenomenon is known as redundancy). On the other hand, if the delay time is quite large, then due to the sensitivity of the chaotic movement, the coordinates appear to be independent and the reconstructed phase space

looks random or complex (a phenomenon known as irrelevance) (Bradley and Kantz, 2015).

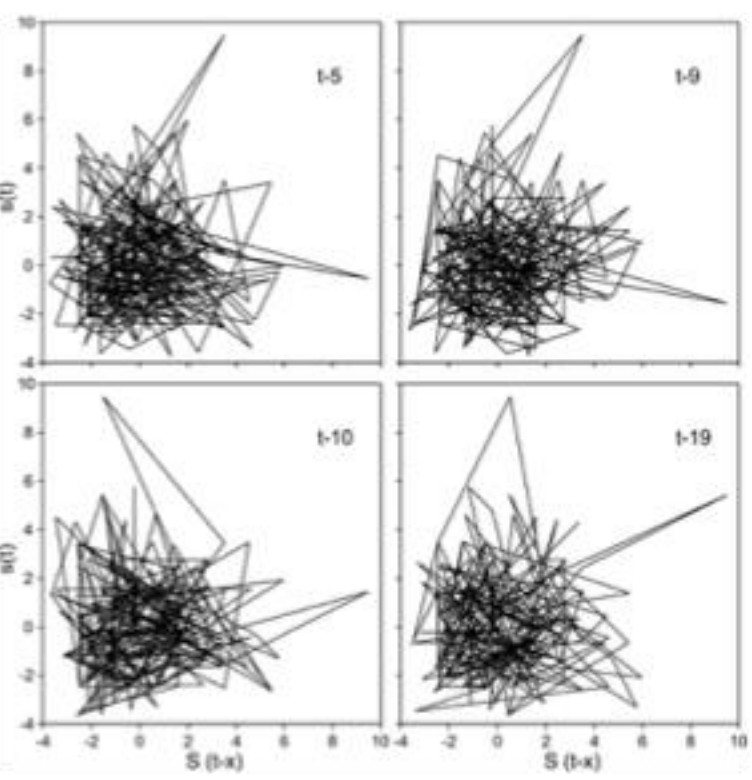

Figure 2. Phase diagrams corresponding to the time series of hurricanes that occurred between the
years 1749 and 2012 in the Gulf of Mexico and the Caribbean Sea. The $x$ - axis in the four plots
indicates the time lag ($\tau$).

The Hurst exponent helps us to identify the criteria to find a time lag, and it also describes the
system behavior (Quintero and Delgado, 2011). This could indicate that the system does not have
chaotic behavior; however, the remaining methods have indicated the opposite, and as mentioned
previously, the Lyapunov exponent is considered the most appropriate method for this type of
dataset. Therefore, different methods will provide different results, but the time series will indicate
the best method and the result we should use.
It was possible to observe the difference in the time lag obtained through the autocorrelation
function and the mutual information; however, it is necessary to use only one result. Through the
space-time separation graphic and the False Close Neighbors method, we obtained embedding
dimensions of $m = 4$ for a $\tau = 9$ and $m = 5$ for $\tau = 10$, and the Theiler window with a value of $W =$
16 for $\tau = 9$ and $W = 18$ for $\tau = 10$ (Fig. 4). The choice of this window is very important so as not
to obtain subsequent spurious dimensions in the attractor. According to Bradley and Kantz (2015),
the Theiler window ensures that the time spacing between the potential pairs of points is large
enough to represent a distributed sample identically and independently.
The idea of the False Close Neighbors algorithm is that at each point in the time series, $\bar{S}_t$ and its
neighbor $\bar{S}_J$ should be searched in a $m$-dimensional space. Thus, the distance $\|S_t - S_j\|$ is
calculated iterating both points, given by:

$$R_i = \frac{S_{i+1} - S_{j+1}}{\|\bar{S}_l - \bar{S}_J\|} \tag{1}$$



If $R_i$ is greater than the threshold given by $R_t$, then $S_J$ has false close neighbors. According to
Kennel et al. (1992), a value of $R_t = 10$ has proven to be a good choice for most of the data set, but
a formal mathematical proof for this conclusion is not known; therefore, if this value does not give
convincing results, it is advisable to repeat the calculations for several $R_t$ (Perc, 2006). In our case,
this value gave relevant results. It may have some False Close Neighbors even when working with
the correct embedding dimension. The result of this analysis may depend on the time lag (Kantz
and Schreiber, 2004). In the same way as the delay time, the value of the embedment dimension is
crucial not only for the reconstruction of the phase space but also to obtain the Lyapunov
exponent. Choosing a large value of $m$ for chaotic data will add redundancy and will affect the
development of many algorithms such as the Lyapunov exponent (Kantz and Schreiber, 2004).
The Lyapunov ($\lambda$) exponents were obtained using the Kantz and Rosenstein methods and took the
time lag, the embedding dimension and the Theiler window as the main values; nevertheless, an
election of the neighborhood radius for the exploration of trajectories was also made, as well as
the points of reference and the neighbors near these points. The modification of these parameters
is important to corroborate the invariant characteristic of the Lyapunov exponent. The Kantz
(1994) method using a value of $m = 4$ and $\tau = 9$ gave us an exponent of $\lambda = 0.483$, while for $m = 5$
and $\tau = 10$ the exponent was $\lambda = 0.483$. Since $\lambda$ is a positive value, it was inferred that our system
is chaotic. In addition, the value of $\lambda$ obtained for both imbibing dimensions was the same,
suggesting that our result is accurate. Using the Rosenstein et al. (1993) method, the value
obtained for $m = 4$ and $\tau = 9$ was $\lambda = 0.1056$, and for $m = 5$ and $\tau = 10$, the exponent was $\lambda =$
0.112 (Fig. 5).
There was a difference between placing the attractor in an embedding dimension of $m = 4$ and one
of $m = 5$; a better unfolding of the attractor in the embedding dimension was observed in $m = 4$
and $\tau = 9$. This value of $\tau$ was obtained with the mutual information method, which, according to
Fraser and Swinney (1986) and Krakovská et al. (2015), provides a better criterion for the choice
of delay time than the value obtained by the autocorrelation function.
It was possible to obtain the Correlation Dimension $D_2$ (Fig. 6) and the Correlation Integral (Fig.
6) using the embedding dimension, the delay time and the Theiler window, following the method
of Grassberger and Procaccia (1983a, 1983b). This was done in order to obtain the possible
dimensions of the attractor. It should be noted that there is a whole family of fractal dimensions,
which are usually known as Renyi dimensions, but these are based on the direct application of box
counting methods, which demands significant memory and processing and whose results can be
very sensitive to the length of the data (Bradley and Kantz, 2015). That is why we use the
Dimension and Integral Correlation, since according to Bradley and Kantz (2015) it is a more
efficient and robust estimator.

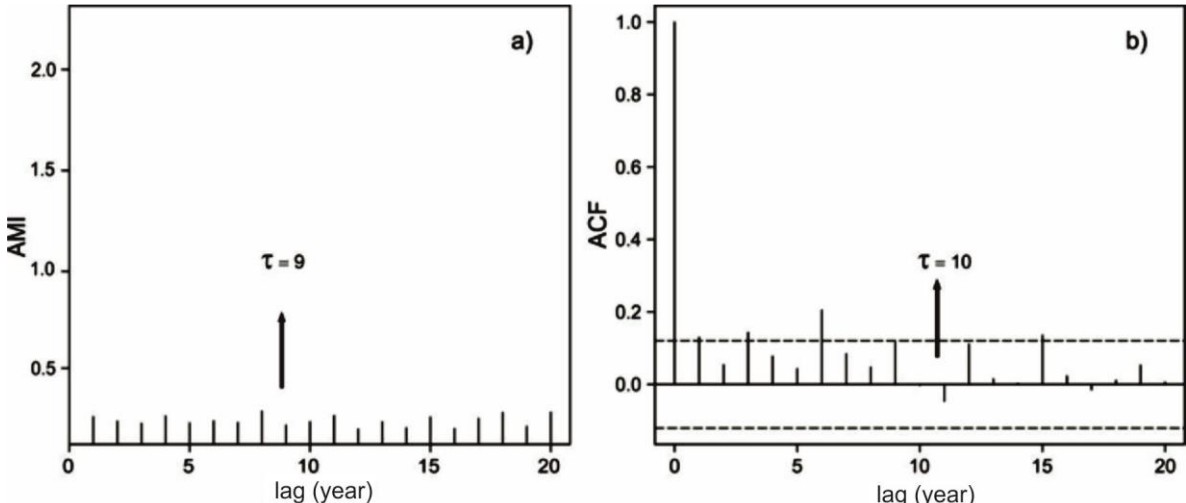

Figure 3. The left panel shows the mutual information method; the $x$ - axis indicates the time lag
against the mutual information index (AMI) and the arrow indicates the first, most pronounced
minimum with a value of $\tau = 9$. The right panel shows the autocorrelation function, the $x$ - axis
indicates the time lag versus the value of the autocorrelation function, and the arrow denotes
where the first zero of the function $\tau = 10$ was obtained.

The right panel on Fig. 7 shows the slope trend of the majority of the slopes of the Correlation
Integral ($\varepsilon$). In the range of $1 < \varepsilon < 10$, we are required to have straight lines as an indicator of the
self-similar geometry. The value obtained here corresponds to $D_2 = 2.20$ which is the
aforementioned slope value. Another method to see the attractor dimension is the Kaplan-Yorke
Dimension ($D_{ky}$), which is associated with the spectrum of Lyapunov exponents and is given by:

$$D_{KY} = k + \sum_{i=1}^{k} \frac{\lambda_i}{|\lambda_{k+1}|}$$ (2)

where $k$ is the maximal integer, such that the sum of the $k$ major exponents is not negative. The
fractal dimension with this method yielded a value of $D_{ky} = 2.26$, which is similar to the one
obtained previously.
Even when all the requirements necessary to apply the nonlinear analysis to our time series are
present, one final requirement must be fulfilled to know whether we can obtain a dimension and
whether the complete spectrum of Lyapunov exponents (another method to visualize chaos) still
needs to be employed.

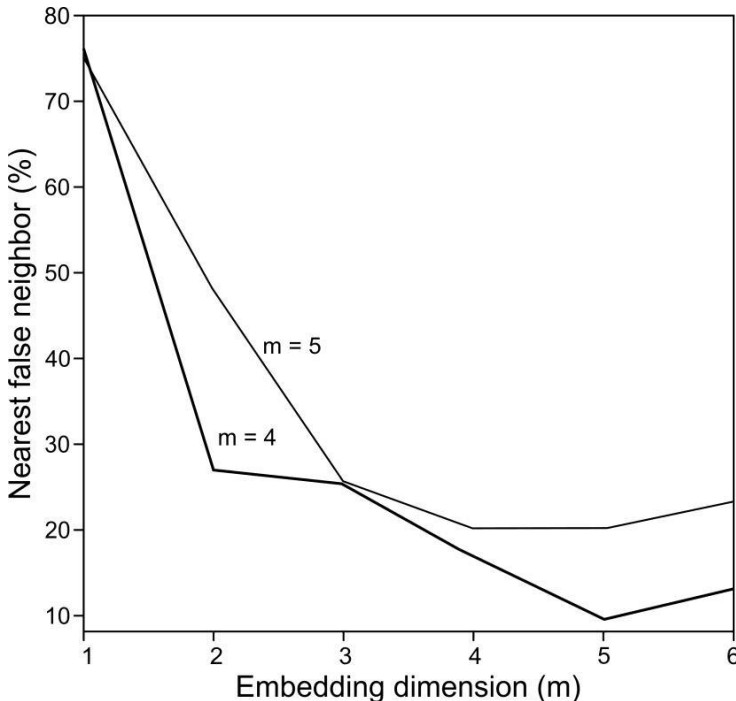


Figure 4. False Close Neighbors with a time lag of 10, where the embedding dimension of 5 has a
9.4% and the embedding dimension of 4 has a 16.66% False Close Neighbors (lower line). False
Close Neighbors with a time lag of 9, where the embedding dimension of 5 has a 20.15% and the
embedding dimension of 4 has a 20.12% False Close Neighbors (upper line). The values in each
line indicate the optimal dimension for each lag.

Eckmann and Ruelle (1992) discuss the size of the dataset required to estimate Lyapunov
dimensions and exponents. When these dimensions and exponents measure the divergence rate
with near-initial conditions, they require a number of neighbors for a given reference point. These
neighbors may be within a sphere of radius $r$ and of a given diameter ($d$) of the reconstructed
attractor.

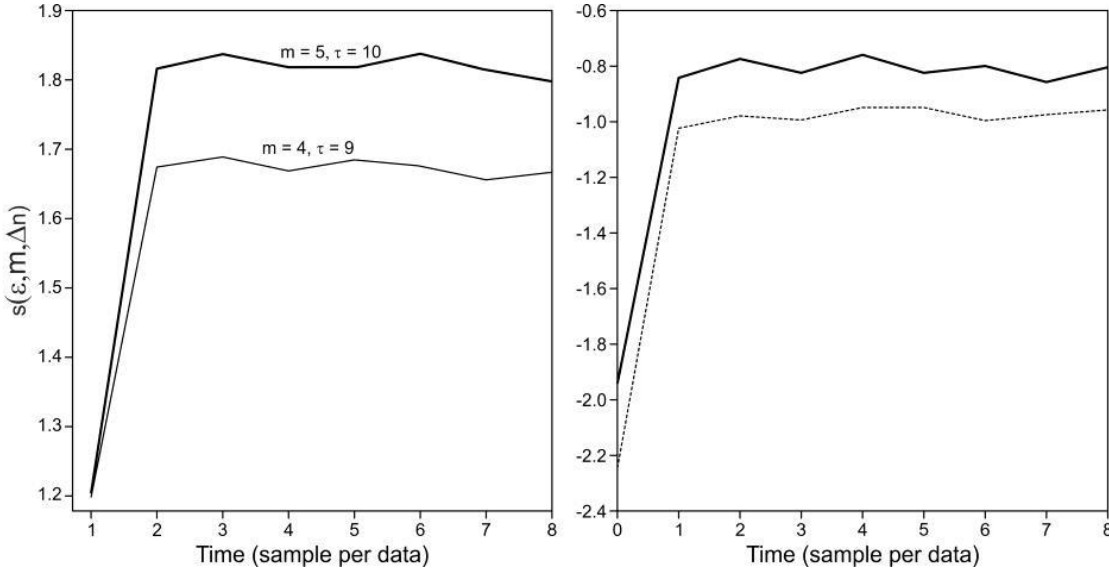


Figure 5. Left panel: Lyapunov exponent with $m = 4$, $\tau = 9$ and $m = 5$, $\tau = 10$, with the Kantz

method. Right panel: Lyapunov exponent with the same values with the Rosenstein method.


In this way we have the requirement for the Eckmann and Ruelle (1992) condition to obtain the

Lyapunov exponents as:


$$logN > Dlog \left(\frac{1}{\rho}\right) \qquad (3)$$


where $D$ is the dimension of the attractor, $N$ is the number of data points and $\frac{r}{d} = \rho$. For $\rho = 0.1$ in

equation (3), $N$ may be chosen such that:


$$N > 10^D \qquad (4)$$


Our time series met this requirement; therefore, it supports our previous results.

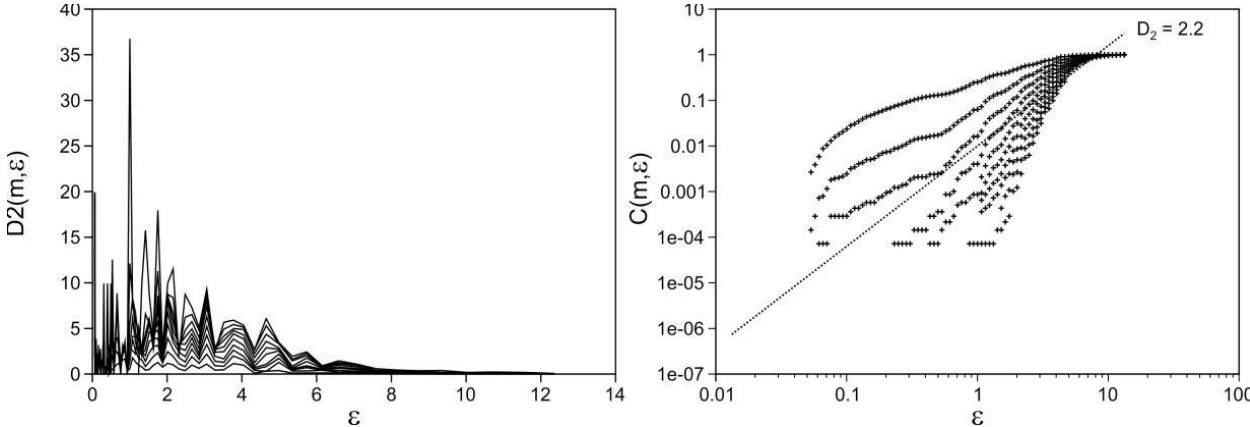


Figure 6. The correlation dimension $D_2$ corresponding to the occurrence of hurricanes in the years
1749-2012 in the Gulf of Mexico and the Caribbean Sea. Left-panel: curves for different
dimensions of the attractor (*y* - axis). Right panel: same information for the $D_2$ with a logarithmic
scale on the *x* - axis.

The attractor dimension was mainly obtained because this value tells us the number of parameters
or degrees of freedom necessary to control or understand the temporal evolution of our system in
the phase space and helps us to know how chaotic our system is. Using the previous methods, a
final fractal dimension of $D_2$ =2.2 was obtained. Following the embedding laws, it must be that *m*
> $D_2$ (Sauer and Yorke, 1993; Kantz and Schreiber, 2004; Bradley and Kantz, 2015). The criterion
of Ruelle (1990) was used to corroborate that the obtained dimension of the attractor is reliable,
where it must be that $N = 10^{\frac{D_2}{2}}$; once the data fulfill this requirement, we can say that the
dimension of the attractor is reliable. Finally, the results indicated that at least three parameters are
needed to characterize our system, since the 2.2 dimension indicates that the attractor dimension
falls between 2 and 3.
The spectrum of the Lyapunov exponent gives 0.09983, -0.07443, -0.23387 and -0.73958;
therefore, the total sum was $\lambda_i = -0.9480$, and according to the previous theory, it is enough have
at least one positive exponent in the spectrum of our system in order to have chaotic behavior.
Finally, the total sum of the spectrum of Lyapunov exponents was negative, indicating that there is
a stable attractor, as mentioned previously. However, since the stable attractor was not easily
distinguished, we used a final method in order to confirm if our system presented some chaotic
dynamic behavior. This method comprised the Iterated Functions System test (IFS) (Fig. 7).
Using Fig. 7, it can be observed that the points representing our system occupy the entire space;
according to the IFS test, there are two possible explanations: 1) The distribution belongs to a
white noise signal and in systems without experimental noise, the point distribution gives a single
curve (Jensen et al., 1985). However, the previous Hurst exponent obtained was not equal to zero;
thus, the white noise was also discarded with the autocorrelation function. 2) The system is chaotic
of high dimensionality. So far, our results have converged on the occurrence of hurricanes in the
Gulf of Mexico and the Caribbean Sea being a chaotic system, so it is feasible to adopt the second
explanation. On the other hand, our Lyapunov exponent figure was not flat and it did not seem to
flatten as the dimension of embedding increased, which, according to Rosenstein et al. (1993),
would mean that our system is not chaotic. Similarly, the Lyapunov exponent increased with the
decrease in the embedment dimension, which is, again, a characteristic of chaotic systems. It was
then also possible to obtain a dimension of the attractor and a positive Lyapunov exponent.
Our results were not easy to interpret because the series presented certain periodic characteristics
in an oscillatory fashion and chaotic behavior at the same time. According to Rojo-Garibaldi et al.
(2016), the series of hurricanes with the spectral analyzes carried out presented strong periodicities
that correspond to sunspots, which gives the system the periodic behavior mentioned above.
According to Zeng et al. (1990), the spectral power analysis is often used to distinguish a chaotic
or quasi-periodic behavior of periodic structures and to identify different periods embedded in a
chaotic signal. Although, as Schuster (1988) and Tsonis (1992) mention, the power spectrum is
not only characteristic of a process of deterministic chaos but also of a linear stochastic process. In
our case, this behavior was not observed in the spectra obtained, which allowed us to detect
periodic signals. The spectra give our system two types of behavior. First, there are periodic
behaviors associated with external forcing, such as the sunspot cycle, giving the system sufficient
order to develop; on the other hand, external forcing presents a chaotic behavior, which gives the
system a certain disorder to be able to adapt to new changes and evolve. The IFS test showed that
the occurrence of hurricanes in the Gulf of Mexico and the Caribbean Sea is chaotic with high
dimensionality. Fraedrich and Leslie (1989) analyzed the trajectories of hurricanes in the region of
Australia and calculated the dimensionality of this process, obtaining a result of between 6 and 8,
i.e., a chaotic process of high dimensionality, which is similar to what we find with the IFS
method. On the other hand, Halsey and Jensen (2004) postulate that hurricanes contain a large
number of dimensions in phase space.
One possible explanation is localized within a boundary where chaos and order are separated; this
boundary is commonly known as the "edge of chaos" (Langton, 1990; Miramontes et al., 2001).
Miramontes et al. (2001) found this type of behavior in ants of the genus Leptothorax, when
studying them individually and in groups. In the former case, the behavior was periodic, while in
the latter, the behavior was chaotic. In our case, we believe that the chaotic behavior is due to the
individual behavior or the hurricane category, since the high dimensionality suggested by the IFS
test agrees with the high dimensionality reported by Fraedrich and Leslie (1989) obtained by
studying the trajectories of hurricanes, that is, by studying them individually, while the periodic
response is due to the behavior of hurricanes as a whole.

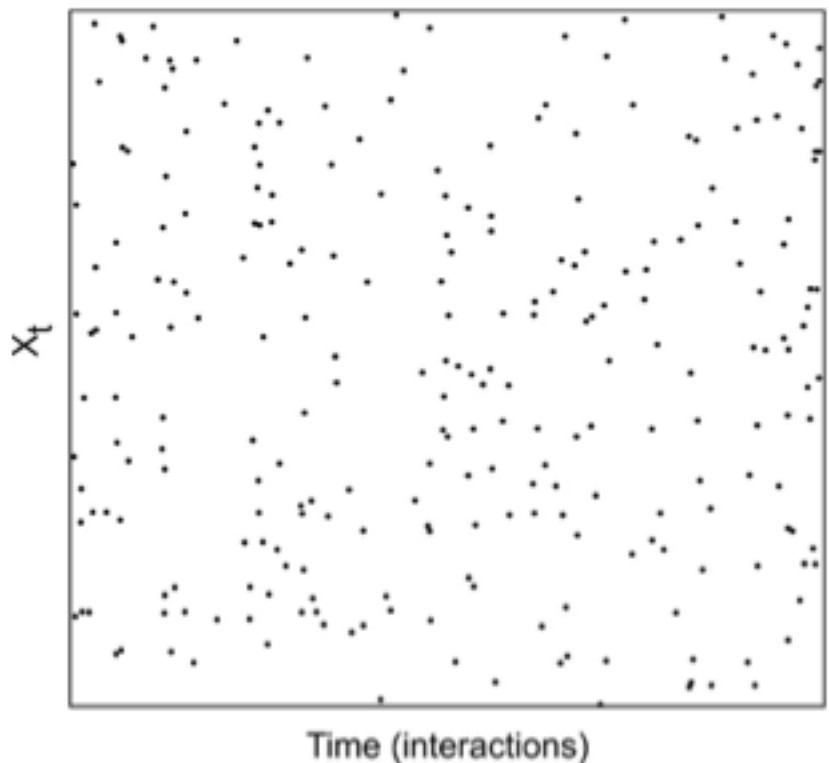

Time (interactions)

Figure 7. The Iterated Functions System (IFS) test applied to the time series of the number of
hurricanes that occurred in the Gulf of Mexico and the Caribbean Sea between the years 1749 and

348 2012.

On the other hand, the entropy test was performed using the non-linear methods and the locally
linear prediction (making the prediction at one step), both methods showed a predictability value
of 2.78 years. The locally linear prediction method was applied as follows: The last known state of
the system, represented by a vector $x = [x(n), x(n + \tau), ..., x(n + (m-1)\tau)]$, is determined, where $m$
is the embedment dimension and $\tau$ is the delay time. Then we found the $p$ nearby states (usually
close neighbors of $x$) of the system that represents what has happened in the past, this was
obtained by calculating their distances to $x$. The idea is then to adjust a map that extrapolates $x$ and
its neighboring $p$ to determine the following values (Dasan *et al.*, 2002). Based on the above, the
value of the embedding dimension and the delay time were changed. Different values of $m$ were
used in order to elucidate the most accurate result; this was obtained with a dimension of $m = 4$
and $\tau = 9$, which are the values for which the attractor of the system was obtained. Therefore, a
good prediction is possible until $t = t_0 + 3$ (Fig. 8). Further if we get away from the measured data
(reported hurricanes), the uncertainty grows in an oscillatory way. For the first two data (2013 and
2014), the absolute error in the prediction (observed value - predicted) is less than 0.2, for the third
and fourth value (2015 and 2016) it is between 0.2 and 0.3 and for 2017 the error is much greater
and gives an overestimation of one hurricane (Figure 8). An important result of this studies is that
they allow to establish the predictability range of a system.

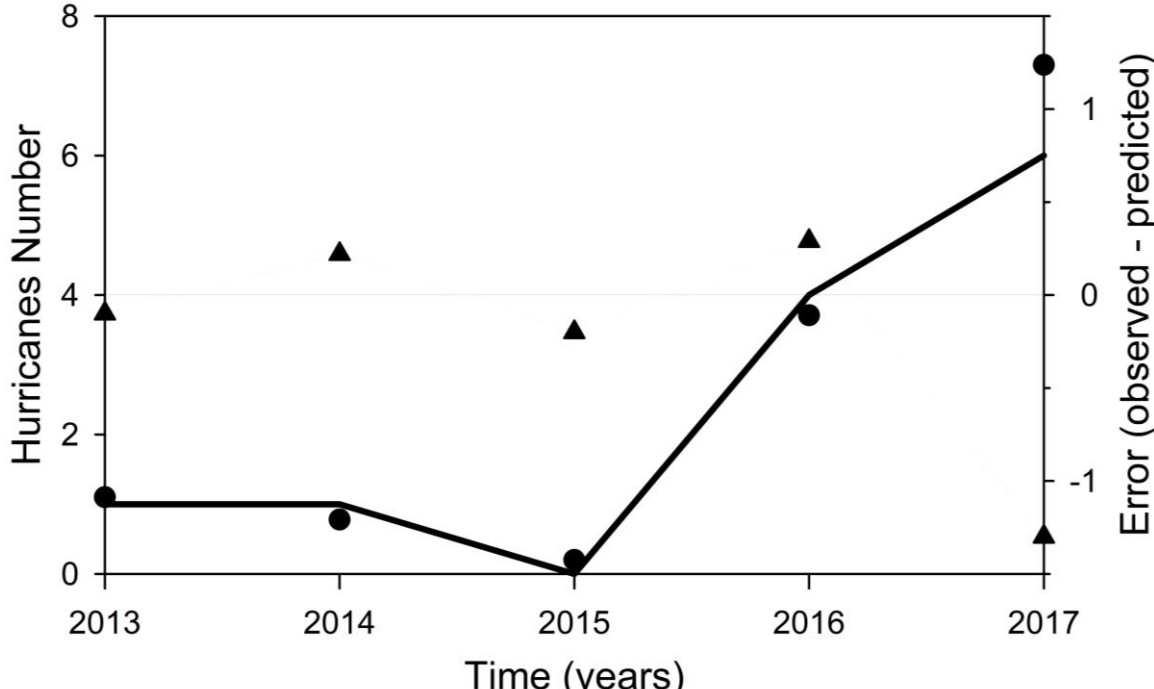


Figure 8. Prediction of the hurricanes number in the Gulf of Mexico and the Caribbean Sea by
means of non-linear methods, and the entropy test. The solid black line represents the number of
hurricanes observed from 2013 to 2017. The black points are the prediction and the triangles is the
error in the prediction considered as the observed value minus the predicted value.

**4 Conclusions**
The results obtained with the nonlinear analysis suggested a chaotic behavior in our system,
mainly based on the Lyapunov exponents and correlation dimension, among others. However, the
Hurst exponent indicated that our system did not follow a chaotic behavior, and in order to be able
to corroborate our results, we employed the IFS method, which led us to think that the hurricane
time series in the Gulf of Mexico and the Caribbean Sea from 1749 to 2012 had a chaotic edge. It
is important to emphasize that this study was prepared as an attempt to understand the behavior of
the occurrence of hurricanes from a historical perspective, since this type of phenomenon is part of
an ocean-atmosphere interaction that has been changing over time, hence the value of our
contribution. However, we are aware that from the time the study was conducted to the present
date there are new records, which will make it possible to carry out new studies applying new
methods.

*Author contributions*. All the authors contributed equally to this work.
*Competing interests*. The authors declare that they have no conflicts of interest.
*Acknowledgements*. This work was financially supported by the Instituto de Ciencias del Mar y
Limnología de la Universidad Nacional Autónoma de México, projects 144 and 145. BR-G is
grateful for the CONACYT Scholarship that supported her study at the Posgrado en Ciencias del
Mar y Limnología, Universidad Nacional Autónoma de México.

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
