# Peer review of "Nonlinear analysis of the occurrence of hurricanes in the Gulf of"

_Nonlinear Processes in Geophysics, 2017_

## Referee Comment (RC1) · Anonymous Referee #1 · 31 Oct 2017

The manuscript's goal is to evaluate the chaotic nature of a hurricane time series reconstructed using historical data and a hurricane dataset. This manuscript presents several problems that I will try to elucidate here:

1) Conception of the study: -it is not clear how the dataset analyzed is constructed and what are the relations with the HURDAT database. No comparison with already-existing hurricane datasets is shown.

-Figure 1: is the linear fit showing a significant reduction in the number of hurricanes? If the fit is significant, this means that your analysis cannot be performed because the series would be issued from a non-stationary process.

-Why the authors perform a nonlinear time series analysis on the time series on the number of hurricanes? What is supposed to be the ubderlying "dynamical system" that generate the hurricanes count in certain regions? How can the "attractor" of the number of hurricane occurrences give any information on the predictability of the phenomena as claimed at the end of the introduction?

-The phase space reconstructions in Figure 2 shows a noisy fixed point structure. This is coherent with the fact that the hurricane occurrence seem to be Poisson or Compound Poisson distributed. The rest of the analysis just show trivially the consequences of this.

2) The language: I won't comment here on the English language but only on the use of scientifically wrong expressions. Just to make some example:

- Hurricanes are not complex systems. They are extreme phenomena that occur in a complex dynamical systems (the climate system).

- Line 7-8 chaotic edge of what?

- The last sentence seems broken or is impossible to understand what you mean by "category". Do you mean hurricane strength?

- "Lyapunov exponent is a key point": actually it is a dynamical systems metric.

- What is a "chaotic movement"?

3) The references are not updated: they are mostly coming from the (excellent) scientific knowledge of dynamical systems in the 80s/90s. There are only very few references from 2000. Of course, since this date there have been several improvements to the methodology and the problems the authors want to address but they seem completely unaware of this body of literature.

4) The conclusions are practically inexistent (this problem is certainly related to the wrong conception of this study, as detailed in my point 1)

For all these reasons, I firmly advise against publication of this manuscript in NPG. I encourage the authors 1) to analyze different time series than the number of hurricanes in the Caribbean region to infer dynamical properties, 2) to review the recent literature on dynamical systems metrics 3) To use carefully the scientific jargon pertinent of dynamical systems community.

---

## Author Comment (AC1) · 29 Nov 2017

The manuscript's goal is to evaluate the chaotic nature of a hurricane time series reconstructed using historical data and a hurricane dataset. This manuscript presents several problems that I will try to elucidate here: 1) Conception of the study: -it is not clear how the dataset analyzed is constructed and what are the relations with the

[Figure]

HURDAT database. No comparison with already existing hurricane datasets is shown.

In the article by Rojo-Garibaldi (2017), the way in which the time series of the hurricane was constructed is explained in detail. The series was elaborated taking the existing information from different hurricane databases. HURDAT is a hurricane database of the NOAA with data from 1851 to 2016. All hurricane from the Gulf of Mexico and the Caribbean Sea were extracted from these dataset. Data prior to HURDAT were obtained from various databases and were considered valid if they were mentioned in at least two different databases. Since the series was built with data from these series, it is not valid to compare the constructed series with the same HURDAT series.

-Figure 1: is the linear fit showing a significant reduction in the number of hurricanes? If the fit is significant, this means that your analysis cannot be performed because the series would be issued from a non-stationary process.

Yes, the linear adjustment shows a reduction in the number of hurricanes. Previous results regarding the number of hurricanes shows an increase in the number, but these studies were conducted with shorter series, the same interval of time used in those studies was analyzed here, and the results showed the same pattern as those reported by those authors, once the time series was extends the trend changes, this is one of the results obtained by Rojo-Garibaldi et al. (2017). In our analysis we start by considering that the time series analyzed is generated by some dynamic system, that has a finite time horizon, that is, for large time series the correlations between the states of the system fall exponentially. It must be taken into account that this requirement may not be valid for critical dynamic systems, which are characterized by dynamic correlations and statistics that decay very slowly, as an inverse power of time. When working with stationary dynamic processes, it does not make sense to take the temporal values in a range higher than the finite time horizon of the system, that is, the one above which there is no correlation between the values of the dynamics. If the adjustment is significant, the series may not be stationary, but other tests are required to be certain and that is why several tests were applied, not only the adjustment. The results show

that it is possible to do the nonlinear analysis and that the results obtained here are valid.

-Why the authors perform a nonlinear time series analysis on the time series on the number of hurricanes?

This is a very interesting question, because it was the main reason for this study and the answer is very simple, because we are interested in model the generative process that gives rise to the complex system. In this case, the question would be: what is the simplest model that is able to explain the observed data?

What is supposed to be the ubderlying "dynamical system" that generate the hurricanes count in certain regions?

This is the key question of most of the studies related to predict the number of hurricanes and its intensity from one year to the next. DeMaria (2008) in the introduction of his article "A Simplified Dynamical System for Tropical Cyclone Intensity Prediction" makes an excellent presentation of the problems and methods used to treat this problem. We start from a different approach, we try to find the nonlinear properties of the system and establish the minimum number of variables required to construct the simplest model that can explain the behavior of the observed data.

How can the "attractor" of the number of hurricane occurrences give any information on the predictability of the phenomena as claimed at the end of the introduction?

As mentioned in the manuscript "The attractor dimension was mainly obtained because this value tells us the number of parameters or degrees of freedom that are necessary to control or understand the temporal evolution of our system in the phase space, and helps us to know how chaotic our system is. . . "

-The phase space reconstructions in Figure 2 shows a noisy fixed point structure. This is coherent with the fact that the hurricane occurrence seem to be Poisson or Compound Poisson distributed.

As mentioned in the manuscript "We observed that the points are scattered in the constructed plane, indicating that there is a chaotic behavior", but we also say "However, the most robust method to identify chaos within the system is the Lyapunov exponent". The Lyapunov (A) exponents were obtained using the Kantz and Rosenstein methods taking the time lag and the embedding dimension. The Kantz (1994) method using a value of m = 4 and r = 9 give us an exponent of A = 0: 48392 and form = 5 and r = 10 the exponent was of A = 0: 48392. Since A is a positive value, it was inferred that our system is chaotic. In addition, the value of A for both imbibing dimensions was the same, suggesting that our result is accurate.

In this manuscript several methods were used, which seems to be redundant, this was because as the referee says, "the hurricane occurrence seems to be Poisson or Compound Poisson distributed", this part "seems to be" requires a very careful analysis.

The rest of the analysis just show trivially the consequences of this. 2) The language: I won't comment here on the English language but only on the use of scientifically wrong expressions. Just to make some example: - Hurricanes are not complex systems. They are extreme phenomena that occur in a complex dynamical systems (the climate system).

This comment reflects how complex and complicated the topic is. Hurricanes, in fact, are extreme events that occur in a complex dynamic system (the climate system) and, in turn, hurricanes are dynamically complex systems, both are true and both have different behaviors from a non-linear point of view. That is why it is possible to predict the trajectory of hurricanes with a good approximation, but the number and intensity of hurricanes that will occur from one year to the next is a more complex problem.

- Line 7-8 chaotic edge of what?

The concept of chaotic edge is relatively new and applies to systems in which an individual has a chaotic behavior while the sets of individuals of the same species have a different behavior, as is the case of ants, bees and, in our results the case of hurri-

canes.

- The last sentence seems broken or is impossible to understand what you mean by "category". Do you mean hurricane strength?

The intensity of hurricanes is measured by the Saffir-Simpson index and is a scale that measures the intensity (or category) of hurricanes based on wind speed.

- "Lyapunov exponent is a key point": actually it is a dynamical systems metric.

The fact that it is considered a dynamical systems metric it does not take away the importance, because as it is said in the manuscript "The Lyapunov exponent is invariant under soft transformations, because it describes the long-term behavior, providing an objective characterization of the corresponding dynamics (Kantz and Schreiber, 2004). The presence of chaos in dynamic systems can be solved by this exponent, since it quantifies the exponential convergence or divergence of initially close to the state space and estimates the amount of chaos in a system (Rosenstein et al., 1993; Haken, 1981 Wolf, 1986)... "

- What is a "chaotic movement"?

The concept of chaotic movement is widely discussed in the specialized literature and it is not our intention to enter into a wide discussion, we will only say that a chaotic system is entirely deterministic, while a random system is completely non-deterministic

3) The references are not updated: they are mostly coming from the (excellent) scientiïfic knowledge of dynamical systems in the 80s/90s. There are only very few references from 2000. Of course, since this date there have been several improvements to the methodology and the problems the authors want to address but they seem completely unaware of this body of literature.

The referee is right and the observation is appreciated. Most of the literature refers to the basic definition of the applied methodologies since the existing one regarding hurricanes refers mostly to the study of the non-linear dynamics of training and the

evolution of hurricanes and not to the nonlinear analysis of time series of the occurrence of hurricanes, however, there is literature that can be used, although it does not refer explicitly to the problem in question.

4) The conclusions are practically inexistent (this problem is certainly related to the wrong conception of this study, as detailed in my point 1)

The most important conclusion that we obtain and, which is the one that is reported in the manuscript, is that the time series of the number of hurricanes shows that the system is in a chaotic edge. This is one of the reasons why it is so difficult to predict the number of hurricanes over time. Future studies on this subject should be done considering this condition.

For all these reasons, I firmly advise against publication nof this manuscript in NPG. I encourage the authors 1) to analyze different time series than the number of hurricanes in the Caribbean region to infer dynamical properties, 2) to review the recent literature on dynamical systems metrics 3) To use carefully the scientific jargon pertinent of dynamical systems community.

We appreciate all the comments, which we will use to improve the manuscript.
* * *

---

## Referee Comment (RC2) · Anonymous Referee #2 · 27 Dec 2017

The paper by Rojo-Garibaldi et al. shows a nonlinear time-series analysis of yearly occurrence hurricane data in the Gulf of Mexico. Yearly data from 1749 to 2012 were used to analyze the deterministic nature of a possible chaotic attractor underlying the time series. To that end, the embedding dimension, time delay, maximum Lyapunov exponents, correlation dimension, etc were calculated using well known techniques.

Concerning the manuscript itself, I am not a native English speaker, but in my opinion the English needs some revision. The Results section is difficult to follow. The amount of information is huge and sometimes it seems that parameters are calculated without rhyme or reason.

[Figure]

The paper needs a major revision and cannot be accepted for publication in NPG in its present form. Find below my major concerns and minor comments.

Major items:

Data set description. I found this section quite discouraging as no information on how the time series was built. For example, are you using yearly data? Only landfall hurricane data were used?, If this is not the case, which is the area for hurricane detection? How Fernández et al data were added to HURDAT data base?

So, in the following I will assume that you are using yearly data since 1749 to 2012 (264 data points).

One of my main concerns is that you are using a time series with some trend. You need to show the stationarity of the series. There is a significant body of literature on nonstationary tests that often deals with the division of the time series into several windows where a common statistical property is measured and compared among the different divided parts of the series. You may plot the probability density function for half of the time series and for the full pack, or you may detrend the time series, for example.

The order of figures seems to me quite confusing. For example, the Poincaré map is shown before the embedding dimension is calculated. Taking into account that m is larger than 3, the section obviously does not give any information at all. It is just a cloud of points, but not due to the chaotic nature of the attractor but as it is a 2D projection of an m-1 dimension set.

The obtained embedding dimension m=4 or 5 is said that correspond to a fractal dimension D=2.2 (m=2D). This corresponds a minimum number of data points (Eq.(9)) N>159 which is more than half of the total length of the data set used in this paper. Other authors (Bountis et al. 1993) use a different formula m=2D+1 and N>10**(2+0.4D) which extends the minimum number of data points beyond the actual length of your time

series. In resume, your data set is too small for this kind of nonlinear analysis.

Figure(7) should appear before than others, and the slope should saturate for the embedding dimension calculated previously (m=4,5) which is not the case here. Any explanation? What's the difference with Fig.(9)?

In Fig.(9) I cannot understand how the linear fitting is done. Why the lines drawn in panel (a) do not match the slopes marked with points. The saturated values on top of the figures should not be takin into account.

The largest Lyapunov exponent as a function of the embedding dimension m and the time delay tau may show the deterministic nature of the time series. When the dimensionality of the embedding space is reduced, the exponent is expected to increase for a deterministic system because the attractor occupies a larger portion of the available space, which does not happen for a random signal. You may compare your results with a random time series (just move randomly the data points within the time series). I think this is important as your Lyapunov exponents are quite small.

One of the interesting applications of these nonlinear techniques is their use to forecast the hurricane occurrence. This will allow to (1) know if the attractor reconstruction is correct, and (2) compare this method to classical linear time series analysis as ARMA estimators, for example. The first point should be mandatory.

Minor points:

Line 1 page 2. "...have usually been performed with linear-type analyses..." needs a reference.

HURDAT reanalysis project needs a reference.

References to nonlinear analysis along the paper are very old. Please, look for new ones and show other example where these kinds of techniques were applied.

Some of the explanations given along the text, or even formulae in the results section

are naïve and should not appear or just referenced. Otherwise, you could move some of them to an Appendix for example.

Along the text time delay is written sometimes as r and in other occasions as tau.

Fig.2 is a mess and does not give any information as it is drawn.

Conclusions section is not admissible. You need to resume and explain why your method is better than others and what is new here. You need to strengthen the deterministic nature of your time series if compared to a random signal. If the time series is deterministic, what is the underlying physics behind this fact?

References:

Bountis, T., L. Karakatsanis, G. Papaioannou, and G. Pavlos, 1993: Determinism and noise in surface temperature time series. Ann. Geophys., 11, 947–959

---

## Author Comment (AC2) · 3 Jan 2018

Answers to comments of referee 2

Concerning the manuscript itself, I am not a native English speaker, but in my opinion the English needs some revision. The Results section is difficult to follow. The amount of information is huge and sometimes it seems that parameters are calculated without rhyme or reason

An apology for the English version of the manuscript, the manuscript was reviewed by an American English speaker. Once the modifications suggested by the reviewers

have been satisfactory made, they will sent it to a professional editing system for their correct presentation in English.

Data set description. I found this section quite discouraging as no information on how the time series was built. For example, are you using yearly data? Only landfall hurricane data were used?, If this is not the case, which is the area for hurricane detection? will assume that you cane data were used?, If this is not the case, which is the area for hurricane detection? How Fernandez et al data were added to HURDAT data base? So, in the following I are using yearly data since 1749 to 2012 (264 data points).

The way in which the time series was constructed is explained in Rojo-Garibaldi et al. (2016):

"a historical database (1749-2012) of hurricane occurrences in the Gulf of Mexico and the Caribbean Sea was built. To do this, we started with the HURDAT hurricanes time series and then analyzed historical re- ports of hurricanes provided by ships that passed near the re- ported hurricanes; data from aerial surveys of recent hurricanes beginning in 1944 were also used. This information, plus that of Fernández-Partagás and Díaz (1995a, 1995b, 1996a, 1996b, 1996c, 1997 and 1999), was added to the HURDAT time series to build a new and longer time series of hurricane occurrence in the Gulf of Mexico and the Caribbean Sea. We considered the data valid when the hurricane was reported by more than three different data- bases; we analyzed the path of the hurricanes to avoid counting a hurricane more than once when it was reported in different places with different times but corresponding to the same hurricane."

Regarding how the data of Fernández-Partágas and Díaz were added and processed, it is explained in detail in the HURDAT page, since it published its data along with a series of articles specifying the methodology for each case, since the publication of its database was done by sections of years. The study area is the Gulf of Mexico and the Caribbean Sea, all hurricanes in that region are considered, observed on land or in water. Annual data are being used (264 data), apparently few data, but using the

criteria of Eckmann and Ruelle (1992) and Ruelle (1990). The result showed that they are sufficient for the analysis that was intended to be performed. On the other hand, it is important to mention that it is the longest series of hurricane occurrences in the Gulf of Mexico and the Caribbean Sea, up to this moment.

One of my main concerns is that you are using a time series with some trend. You need to show the stationarity of the series. There is a significant body of literature on nonstationary tests that often deals with the division of the time series into several windows where a common statistical property is measured and compared among the different divided parts of the series. You may plot the probability density function for half of the time series and for the full pack, or you may detrend the time series, for example.

Figure 1 shows the time series and its tendency, in most of the analyzes it is commonly used to use series without trend, like in spectral analysis, non-linear analysis, etc; since leaving the trend gives a residual that masks the results. In our study we removed the trend. We appreciate the observation and we included a paragraph making explicit that the tendency to the series was removed.

The order of figures seems to me quite confusing. For example, the Poincare map is shown before the embedding dimension is calculated. Taking into account that m is larger than 3, the section obviously does not give any information at all. It is just a cloud of points, but not due to the chaotic nature of the attractor but as it is a 2D projection of an m-1 dimension set.

We can change the order of the results and consequently the order of the figures. Our objective was to present the evidences of possible existence of chaos in the series and to apply more robust methods, to corroborate this, as it is said in the following paragraphs:

"The optimal time lag ($\tau$) obtained from Fig. 2 was equal to 9. In our case the hurricane dynamics were not distinguish through with the phase diagram; however, since any hurricane trajectory starts at a close point location on the attractor dataset that diverges exponentially, it is a primary evidence of a chaotic motion according to Thompson and Stewart (1986). Another way to visualize the dynamics of the system is through the Poincare Surface Section (Fig. 3), which helped us to observe the presence of chaos involved in our data. It was observed that the points are scattered in the constructed plane, which indicates that there is a chaotic behavior. However, the most robust method to identify chaos within the system is the exponent of Lyapunov, before obtaining the exponent it was necessary to calculate the time lag using the Theiler window and the embedding dimension"

On the other hand, if the Poincare diagram does not show "a cloud of points" then we can say that there is no chaotic nature. Finally, a two-dimensional scan is the first to say something about the chaotic nature of the data.

The obtained embedding dimension m=4 or 5 is said that correspond to a fractal dimension D=2.2 (m=2D). This corresponds a minimum number of data points (Eq.(9)) N1›159 which is more than half of the total length of the data set used in this paper. Other authors (Bountis et al. 1993) use a different formula m=2D+1 and Nb10**(2+0.4D) which extends the minimum number of data points beyond the actual length of your time series. In resume, your data set is too small for this kind of nonlinear analysis.

In fact, there are other authors that use different formulas or criteria to estimate the minimum number of points that are required to do a non-linear analysis in time series. We use the criteria of Eckmann and Ruelle (1992) and Ruelle (1990), which is well known and used in this type of analysis. Here it is important to mention that it is true that our references are somewhat outdated, but they correspond to quotations from widely used and proven methods, they are not new methods or proposals of novel methods, our objective is not to show that novel methods are used, the main objective is to understand why hurricanes have the behavior that makes them so special and difficult to predict; even with all the technology and theoretical methods used for its detection

and prediction. That's why we applied well-known methods that are not subject to doubts. Regarding a m = 2D, it must be obeyed by the laws of embedding that m > 2Df, but in the case of the correlation dimension, it is sufficient that m > Df (Sauer and Yorke, 1993; Kantz and Schreiber , 2004).

Figure (7) should appear before than others, and the slope should saturate for the embedding dimension calculated previously (m=4,5) which is not the case here. Any explanation? What's the difference with Fig. (9)?

A very pertinent observation, since Figure 9 includes the information of Figure 7, the latter can be removed. With respect to the slope, which must be saturated for the previously calculated dimension (m = 4.5), which apparently is not the case here, the explanation is only for the shape of the figure.

In Fig. (9) I cannot understand how the linear fitting is done. Why the lines drawn in panel (a) do not match the slopes marked with points. The saturated values on top of the figures should not be takin into account.

Figure 9 on the left side shows the exploratory analysis of the possible dimensions of the tractor, which were obtained with the help of figure 8 which is the correlation dimension, in many of the known literature it is known that this only gives an approximate dimension and therefore it is necessary to use the correlation integral, which is figure 9, on the right side we are putting the slope that best fits the dimensions of the attractor. The adjustment of the slope was made according to the method developed by Grassberger and Procaccia (1983a, 1983b). The largest Lyapunov exponent as a function of the embedding dimension m and the time delay tau may show the deterministic nature of the time series. When the dimensionality of the embedding space is reduced, the exponent is expected to increase for a deterministic system because the attractor occupies a larger portion of the available space, which does not happen for a random signal. You may compare your results with a random time series (just move randomly the data points within the time series). I think this is important as your

Lyapunov exponents are quite small.

We appreciate the comment and we did this analysis.

One of the interesting applications of these nonlinear techniques is their use to forecast the hurricane occurrence. This will allow to (1) know if the attractor reconstruction is correct, and (2) compare this method to classical linear time series analysis as ARMA estimators, for example. The first point should be mandatory.

We appreciate the comment. With respect to the first point, different types of analysis were made; in fact, they seem redundant, but it was one of our concerns to have certainty in estimating the attractor. With respect to the second point in Rojo-Garibaldi et al. (2016) a part of the linear and non-linear analysis was done, to the time series applying spectral analysis, we included the reference and a comparison with the results obtained in Rojo-Garibaldi et al. (2016).

Minor points:

We appreciate the comments and we made them punctually throughout the text.

Line 1 page 2. "...have usually been performed with linear-type analyses..." needs a reference.

HURDAT reanalysis project needs a reference.

References to nonlinear analysis along the paper are very old. Please, look for new ones and show other example where these kinds of techniques were applied.

Some of the explanations given along the text, or even formulae in the results section are naïve and should not appear or just referenced. Otherwise, you could move some of them to an Appendix for example.

Along the text time delay is written sometimes as r and in other occasions as tau. Fig.2 is a mess and does not give any information as it is drawn.

Conclusions section is not admissible. You need to resume and explain why your method is better than others and what is new here. You need to strengthen the deterministic nature of your time series if compared to a random signal. If the time series is deterministic, what is the underlying physics behind this fact?

In this case, it is not of our interest to show that the methods that were applied here are better than others, our interest concerns the chaotic nature of the hurricanes. What is important here is to show that hurricanes show a different behavior when they are analyzed in an individual form that when they are studied collectively, that corresponds to what is called state in a chaotic edge. We believe that this is important for future hurricane studies, their individual and collective behavior should be studied separately.

References:

Bountis, T., L. Karakatsanis, G. Papaioannou, and G. Pavlos. 1993: Determinism and noise in surface temperature time series. Ann. Geophys., 11, 947-959
* * *

---

## Author Response (AR1)

Prof. Vicente Perez-Munuzuri

NPG Editor

Dear Prof. Perez-Munuzuri

You will find ci-joint the new version of our manuscript "Nonlinear analysis of the occurrence of hurricanes in the Gulf of Mexico and the Caribbean Sea" by Berenice Rojo-Garibaldi, David Alberto Salas-de-León, María Adela Monreal-Gómez, Norma Leticia Sánchez-Santillán, and David Salas-Monreal. We are sending two versions (tracking) in which you can see the changes made according to the comments of the reviewers and another (new) which is the version with the changes included

New comments for the reviewers:

The other methods used for the nonlinear analysis of the time series of hurricane events in the Gulf of Mexico and the Caribbean Sea, confirmed that this system has a chaotic behavior. When performing the corresponding spectral and finding a periodic behavior at the same time, it can be concluded that the system has a chaotic "edge" behavior.

Different points must be taken into account to validate our results and these are the following:

1. To verify that our system has a chaotic behavior, the exponent of Lyapunov was obtained, but first we draw an embedding dimension and a delay time, this was done with the help of Theiler's window, which according to the chaos theory, it is necessary to not get the spurious dimensions. The exponent of Lyapunov was obtained by two different methods the Kantz (1994) and the Rosenstein (1993) method, in both cases the results were positive, indicating in this way that our system is chaotic.

2. The Poincaré diagram also gave us a chaotic behavior.

3. Since we were aware that our database was short, a bibliographic review was made to know if it was possible to work with this data number. We found that the number of data showed here was in agreement with the number of data expected by Eckmann and Ruelle (1992), therefore the number of data was sufficient to ensure obtaining a reliable Lyapunov exponent (this is discussed in the manuscript).

4. Another way to make sure that our system was chaotic was done through a visual inspection of the Lyapunov exponent graphs, once it approached the dimension of the embedment. According to Rosenstein (1993) the embedment dimension presents a flat behavior in the graph, in order to be not chaotic. This type of behavior was not observed here, on the contrary, the form is maintained (this was also mentioned in the manuscript).

5. For the spectra of the Lyapunov exponent, the following values were obtained: 0.09983, -0.07443, -0.23387 and -0.73958, whose sum has a value of $\Sigma\lambda_i = -0.9480$, according to the theory, it is enough to find at least one positive exponent in the spectrum to prove that our system has a chaotic behavior. Another observation made with respect to the spectrum of the exponents of Lyapunov, is the result obtained by the sum, in this case the value was negative, which indicates that there is a stable attractor.

6. In the case of the dimension of the attractor, we also were sure to get the requirement of Ruelle (1990), in order to have a reliable result of the dimension obtained for the attractor.

7. For the dimension of the attractor, the dimension of Hausdffor and Kaplan-Yorke were obtained, resulting in D2 = 2.2 and Dk-y = 2.26, this suggests that the result obtained for the dimension of the attractor is robust.

8 Finally, the IFS method was used, this suggest that when the points cover the whole space it may indicated two things: 1) The system is a white noise or 2) The system has a chaotic behavior of high dimensionality. As referred in the manuscript, the white noise

was discarded according to the Hurst exponent, therefore it was inferring that the system is chaotic and of high dimensionality.

9. All these analyzes helped to confirm that the system has a chaotic behavior, giving us at the same time more solid foundations for our result. However, and taking into account the suggestion of the reviewers, an additional analysis was carried out to check if our system is indeed chaotic, the exponent of Lyapunov was plotted and its value was obtained as the embedment dimension decreased. This was done at different dimensions (m = 4.3 and 2) and with different radii ($\varepsilon$ = 16, 24 and 30), with this analysis we can highlight two things: 1) The shape of the curve in the exponent of Lyapunov, is keeps invariant before transformations and 2) As suggested by the reviewer, the exponent of Lyapunov increases as the embedment dimension decreases (see figure below).

[Figure]

Where the exponents were:

$\lambda_4=0.0349$, for m=4

$\lambda_3=0.0365$, for m=3

$\lambda_2=0.0480$, for m=2

In this case, the numerical result differs from that obtained previously due to the different value of the radius and the number of iterations used here, however the important result is that the exponent remains invariant and increases its value by decreasing the embedment dimension. We must point out that, however, the values presented in the original document are reliable since the data where carefully chosen and analyzed, in this last analysis the calculations were made only in an exploratory way, in order to corroborate what was suggested by the reviewer.

[revised manuscript text omitted]

---

## Author Response (AR2)

Dear Prof. Vicente Perez-Munuzuri
Editor
Nonlinear Processes in Geophysics

You will find ci-joint the new version of our manuscript "Nonlinear analysis of the
occurrence of hurricanes in the Gulf of Mexico and the Caribbean Sea". We provide a
point-by-point list of each comments by the reviewers. English was significant
improved by the Elsevier Language Editing Services, so we cannot indicate where in
the manuscript the English revision appears. What we do is indicate the changes of the
point-by-point list of each comments by the reviewers (page #, line #).

David Salas

**Answers to the Second Reviewer**

1. *For example, it is not clear for me that the time series is stationary. May you calculate the probability density function for the half and full data lenght and compare the results. Of course after the detrend has been done.*

**Reply:** Two tests were done to see the stationarity of the series, one of which was the Dickey-Fuller test (D-F), which is the standard test to prove the stationarity of a series. They consider three different regression equations that can be used to prove the presence of a unit root, the parameter of interest of these equations is **r**, if **r = 1** the series has a unit root. In this test the null hypothesis where Ho: **r = 1** shows that the series has unit root and is not stationary, and if Ha: **r < 1** then the series is stationary. Using this test, a value of D-F = -5.7753 was obtained with a p-value = 0.01, which is statistically significant and therefore we can say that our series is stationary.

Now, following the test suggested by the reviewer. Since a series is stationary over time when its mean and variance are constant over time, the respective values were obtained for the middle of the series and for the complete series. The values of the mean and variance of the complete series were: mean = 0.138 and variance = 0.020. The values for the middle of the series were: mean = 0.123 and variance = 0.0199, so this requirement is also met. Finally, the probability density function was plotted for 1749-1881 and 1749-2012, as well as their respective histograms, with this we can see how the form of the function and the histogram are conserved over time.

[Figure]

Figure 1. (Left) Histogram from the Hurricane series 1749-2012. (Right) Histogram for the Hurricane series 1749-1881. It is possible to observe how the distribution of the histogram is preserved for the middle of the series and for the complete series.

[Figure]

Figure 2. (Left) Probability density function of the Hurricane series 1749-2012. (Right) Probability density function for the Hurricane series of 1749-1881. It is possible to observe how the distribution of the function is preserved for the middle of the series and for the complete series.

2. *Concerning the Poincaré map I continue to have some doubts as may not be useful. I can think of a 5-dimensional quasiperiodic signal that shows an irregular two-dimensional Poincaré map, and however the original time series is not chaotic I recommend deleting that figure and the corresponding comments.*

   **Reply:** The section corresponding to this part has already been deleted.

3. *The new results of the Lyapunov coefficient as a function of the embedding dimension and the time delay which are shown in the answer do not appear in the manuscript. I think they should.*

   **Reply:** The new results were already put in the text. See page 14, lines 313-315

4. *One of the main objectives of using these nonlinear methods not only lie in obtaining an embedding dimension and showing the chaotic nature of the time series, but to perform some forecasting. What will happen if you try to reconstruct the time series for example, using only half of its length. This was mentioned in my previous review.*

   **Reply:** = By means of non-linear methods, the entropy test was performed, which showed a predictability value of 2.78 years, and means of the locally linear prediction (making the prediction at one step), the same value was obtained. The procedure for this method is as follows: The last known state of the system, represented by a vector $x = [x(n), x(n + \tau), ..., x(n + (m-1) \tau)]$, is determined, where $m$ is the embedment dimension and $\tau$ is the delay time. Then we have found $p$ nearby states (usually close neighbors of $x$) of the system that has happened in the past, from calculating their distances of $x$. The idea is then to adjust a map that extrapolates $x$ and its neighboring $p$ to determine the following values" (Dasan et al., 2002). Based on the above, the value of the embedding dimension and the delay time were changed, in order to see in which values better results were obtained; this
was possible with a dimension of $m = 4$ and $\tau = 9$, which are the values with which
the attractor of the system was obtained. Therefore, a good prediction is possible
until $t = t_0 + 3$. It is not possible to reproduce half of the series since the system tells
us that we can only do it for two years.

Dasan, J., Ramamohan, R. T., Singh, A., y Prabhu, R. N. (2002) Stress fluctuations
in sheared Stokesian suspensions, Phys. Rev., E, 66, pp.021409-1-021409-14.

**Answers to the Third Reviewer**

1. *The manuscript presents nonlinear analysis of the occurrence of hurricanes in the Gulf of Mexico and the Caribbean Sea, which is interesting. The subject addressed is within the scope of the journal.*

2. *However, the manuscript, in its present form, contains several weaknesses. Appropriate revisions to the following points should be undertaken in order to justify recommendation for publication.*

3. *Full names should be shown for all abbreviations in their first occurrence in texts. For example, 2-D in p.6, etc.*

   **Reply**: done

4. *For readers to quickly catch your contribution, it would be better to highlight major difficulties and challenges, and your original achievements to overcome them, in a clearer way in abstract and introduction.*

   **Reply**: Already added to the text. See page 1 y 2, lines 21-22, 40-44

5. *It is shown in the reference list that the authors have a pertinent publication in this field. This raises some concerns regarding the potential overlap with their previous works. The authors should explicitly state the novel contribution of this work, the similarities and the differences of this work with their previous publications.*

   **Reply**: Indeed, we have a publication about the hurricanes that occurred in the same study area for the same time interval (1749-2012). However, in this case, our analysis focused on a different approach, for this article we apply different methods of spectral analysis (Wavelets, Fast Fourier Transform, Multi-taper and Maximum Entropy), in order to see if there was a relationship between the occurrence of hurricanes and the periodicity of sunspots.

   An exhaustive analysis was carried out on the type of correlation between the two systems, as well as the lag between both events. Finding a considerable relationship between the two systems, not only with the periodicities, but also with the type of correlation.

   The substantial difference between the previous article and this one, is that this time we are focusing on the behavior of only the hurricanes, performing a non-linear analysis, the main objective was to find out if the hurricanes belong to the chaotic dynamic systems. When comparing both studies (spectral analysis and non-linear analysis) we realized that hurricanes have the so-called "chaotic edge". This is the most important contribution of our work, since it tells us that hurricanes can behave periodically and follow a pattern, but this does not hold all the time, their behavior goes to a threshold at
which it is chaotic.

**6.**   ***It is mentioned in p.1 that historical records of 1749 to 2012 are taken. Why are***
***more recent data not included in the study?***

**Reply**: This was a study that was carried out during the year of 2012, until then, recent
information was taken.

***6.1. Is there any difficulty in obtaining more recent data?***

**Reply**: Of course not, the HURDAT page updates its data every year after the end of
hurricane season in the Atlantic Ocean, Caribbean Sea and Gulf of Mexico. Well it's
the official hurricanes record of USA for this area.

***6.2. Are there any changes to situation in recent years? What are its effects on the***
***result?***

**Reply**: We must not forget that this type of studies provides us with information about
the dynamics of an underlying system, so that, once we have found that our system is
chaotic, it will not let being chaotic even if we obtain the record of the missing years
(i.e., 2013-2017). Therefore, the absence of this data does not affect our result.

**7.**   ***It is mentioned in p.1 that the Gulf of Mexico and the Caribbean Sea are adopted as***
***the case study. What are other feasible alternatives?***

**Reply**: We can study the hurricanes of the Pacific Ocean, which is another area
impacted by the occurrence of hurricanes, for this part we can analyze those occurred
in the Mexican territory, and those observed in China, Japan and the Philippines. In the
same way, they are observed in Australia and India.

***7.1. What are the advantages of adopting this particular case study over others in this***
***case?***

**Reply**: The importance of this project and the reason why this region was studied, is
because there was a fairly substantial record of the hurricanes that occurred in the Gulf
of Mexico and the Caribbean Sea since 1749. The other regions impacted by hurricanes
do not have such a long historical record. Having a record of this magnitude was
important for the spectral analysis that was developed, on the other hand the length of
the time series, also allowed the development of non-linear analysis, since with less
available data it would then be impossible to think to do this type of analysis.

***7.2. How will this affect the results? The authors should provide more details on this.***

**Reply**: As mentioned above, the study area was chosen taking into account the long
historical record that was available. Even at present, the data in the other regions
impacted by hurricanes, would not be enough to perform a non-linear analysis, at least
for the case in which we want to study the occurrence of these phenomenon.
8.   *It is mentioned in p.1 that HURDAT is adopted as the database. What are other*
*feasible alternatives?*
**Reply**: There is the National Hurricane Center (NHC) -NOAA (National Oceanic and
Atmospheric Administration), which is a regional specialized meteorological center for
the North Atlantic and the Eastern Pacific, created with the purpose of creating a
hurricane warning network. HURDAT considered the annual registration of this center.
*8.1. What are the advantages of adopting this particular database over others in this*
*case? How will this affect the results? The authors should provide more details on*
*this.*
**Reply**: Both HURDAT and NHC-NOAA are official records of the United States and
HURDAT bases its database on the NHC report, apart from its own re-analysis project,
therefore, we consider that our results could not be affected. We have not only reliable
historical data, but also data obtained from the official records of the United States.
9.   *It is mentioned in p.1 that spectral analysis is adopted for the nonlinear analysis of*
*the hurricanes time series. What are other feasible alternatives?*

**Reply**: In fact, the first hint of chaotic behavior can be seen from a spectral analysis.
"Spectral power analysis is often used to distinguish chaotic or quasi-periodic behavior
from periodic structures and to identify different periods embedded in a chaotic signal"
(Zeng et al., 1990). According to Schuster (1988) and Tsonis (1992) the power
spectrum is not only characteristic of a process of deterministic chaos, but also of a
linear stochastic process. That is why more studies should be done and for this reason.
Other than this first approach, we have the Hurst exponent and the phase space graph,
these results are presented in our result section.

*9.1. What are the advantages of adopting this particular approach over others in this*
*case? How will this affect the results? The authors should provide more details on*
*this.*
**Reply**: As we mentioned before, spectral analysis is a tool that is used to see possible
indications of a chaotic behavior, however, it cannot always be appreciated and that is
why the corresponding methods of non-linear analysis are used.

***10. It is mentioned in p.5 that three methods are adopted to know the properties of the***
***system. What are other feasible alternatives?***
**Reply**: There are the bifurcation diagrams which are abrupt changes of the geometry of
the attractor or of the topology in a critical value of the control parameter. In this type
of diagrams, it is possible to see periodic and chaotic regimes, there are five different
types of bifurcation diagrams, which provide a route to chaos. In fact, these types of
diagrams can be compared with the graph of the Lyapunov exponent. For example, for
a logistic map, it can be seen how the Lyapunov exponent goes from negative values in
the regular regions of the bifurcation diagram, to positive values in the chaotic regions,
becoming zero at the bifurcation points. In the chaotic region we can see regular
behavior windows, in which the exponent becomes negative again. On the other hand,
we have the Horizontal Visibility graph method, which "offers a promising new
method for the development of time series analysis, mainly because it has been
corroborated that the fundamental nature of quite different complex dynamic processes
is inherited for the associated visibility charts" (Núñez et al., 2013). The Horizontal
Visibility graphic allow us to describe chaotic, fractal-stochastic and dissipative
processes.
***10.1.      What are the advantages of adopting these particular methods over others***
***in this case? How will this affect the results? The authors should provide more***
***details on this.***
**Reply**: The methods used are as good as those mentioned above. If the Visibility graph
had been used, the advantage would have been not having to calculate the rest of the
parameters such as: delay time, Theiler window, embedding dimension, etc.; however,
we do not use this method because it requires thousands of data.
***11. It is mentioned in p.6 that the algorithms proposed by Kantz (1994) and Rosenstein***
***et al. (1993) are adopted to compute the Lyapunov exponent. What are other feasible***
***alternatives? What are the advantages of adopting these particular algorithms over***
***others in this case? How will this affect the results? The authors should provide***
***more details on this.***
**Reply**: There is the algorithm of Wolf et al. (1985), in which the maximum exponent
of Lyapunov can also be calculated from a data set, following the long-term evolution
of one of the main axes. However, it is a highly sensitive method and can easily lead to
an erroneous result. Rosenstein and Kantz, more than suggesting a trajectory, used the
complete data set and essentially calculated a trajectory for each pair of nearby
neighbors. The two algorithms are substantially similar and calculate the maximum
exponent of Lyapunov by looking for all the neighbors within a neighborhood of the
reference trajectory and calculating the average distance between the neighbors and
that trajectory as a function of time or relative time scale for the data sampling rate.
Having used the algorithm of Wolf et al. (1985) a positive Lyapunov exponent would
have been obtained without guaranteeing that the system has a chaotic dynamic, since
this algorithm always gives a positive exponent.

*12. It is mentioned in p.6 that the Poincaré surface is adopted to detect some kind of*
*chaotic behavior. What are other feasible alternatives? What are the advantages of*
*adopting this particular approach over others in this case? How will this affect the*
*results? The authors should provide more details on this.*
**Reply**: The other alternatives are those presented at the present work; In fact, in our
study we did not rely on the results obtained by a single method, but we used several
methods, in order to corroborate each result.
*13. It is mentioned in p.6 that the "delay method" is adopted to have a qualitative idea of*
*the number of hurricanes that occurred. What are other feasible alternatives? What*
*are the advantages of adopting this particular method over others in this case? How*
*will this affect the results? The authors should provide more details on this.*
**Reply**: The "delay method" was used to construct the phase space of the system, once
the delay time and the embedment dimension were obtained. It was a method created
precisely for the case where we have a discrete system, that is, a set of data; so far it is
the only method.
*14. It is mentioned in p.7 that three different methods are adopted to calculate the time*
*lag. What are other feasible alternatives? What are the advantages of adopting these*
*particular methods over others in this case? How will this affect the results? The*
*authors should provide more details on this.*
**Reply**: The delay time can also be obtained by constructing the phase space from an
arbitrary time delay, later, by trial and error, testing with other values until the
trajectories are more visible; however, this form does not give a very reliable value of
the delay time. That is why we used the mentioned methods, which are, until now, the
most reliable.
*15. It is mentioned in p.10 that the Kaplan-Yorke Dimension is adopted to see the*
*attractor dimension. What are other feasible alternatives? What are the advantages*
*of adopting this particular method over others in this case? How will this affect the*
*results? The authors should provide more details on this.*
**Reply**: There is a whole family of fractal dimensions $D_q$, which are called Renyi
dimensions, the way you can see them is through a partition of the phase space: For the
number of boxes $N_\varepsilon$ of size $\varepsilon$, you need to cover a fractal set with scales of dimension
$D_0$. In $D_0$ we have another type of dimension, which is the so-called capacity
dimension, which is closely related to the dimension of Hausdorff, which is from the
mathematical point of view, the most natural concept to characterize fractal sets. On
the other hand, there is also the Information dimension, which takes into account the
relative frequency of visits of the trajectory, making this type of dimension more
attractive for physical systems. However, the Integral and the Correlation Dimension
were made to characterize measured data, as well as being more robust and efficient
estimators. Successive elements of a time series are not usually independent, but the last two mentioned methods involve phase space vectors such as the location of points
in an attractor. This is why they are the most used and most reliable.
***16. It is mentioned in p.13 that the criterion of Ruelle (1990) is adopted to corroborate***
***that the obtained dimension of the attractor is reliable. What are other feasible***
***alternatives? What are the advantages of adopting this particular criterion over***
***others in this case? How will this affect the results? The authors should provide***
***more details on this.***
**Reply**: We also have the criterion suggested by Tsonis et al. (1993), assuming that
$M \sim 10^{2+0.4\nu_2}$ data points are needed for a reliable estimate of the fractal dimension $\nu_2$.
If this criterion is used, our data does not meet the requirement; however, they agree
with the requirements of Ruelle (1990) and there is no stipulation that requires that the
systems must have both requirements, so it is sufficient for the system to agree with
one of them in order for the dimension to be reliable.
***17. It is mentioned in p.14 that the Iterated Functions System test is adopted to confirm***
***that there is a stable attractor. What are other feasible alternatives? What are the***
***advantages of adopting this particular test over others in this case? How will this***
***affect the results? The authors should provide more details on this.***
**Reply**: The Iterated Functions system is used to make an adequate visualization of fine
details that are present in the time series, including the self-similarity, therefore it can
reveal the correlations in the data and help characterize its "color" (referring to the type
of noise). As for the techniques that are used for the characteristic of the data, we also
have the Hurst exponent, which also characterizes the color of the noise. Both methods
are used in our study. The results obtained from both methods help to complement our
discussion.
***18. It is mentioned in p.15 that "…test showed that the occurrence of hurricanes in the***
***Gulf of Mexico and the Caribbean Sea is chaotic with high dimensionality. One***
***possible explanation is.…" More justification should be furnished on this issue.***
**Reply**: It has already been added to the text, see page 14 y 15, line 316-343.
***19. Some key parameters are not mentioned. The rationale on the choice of the***
***particular set of parameters should be explained with more details. Have the authors***
***experimented with other sets of values? What are the sensitivities of these***
***parameters on the results?***
**Reply**: We do not understand to what key parameters it refers. The parameters that
were used were: 1) The Theiler window, which was obtained from the space-time
separation graph. The value of this window is very important because it prevents
spurious dimensions from being obtained in the attractor. In fact, if a good Theiler
window is not chosen, it is not possible to extract the embedding dimension. 2) On the
other hand we have the delay time and the dimension of embebimiento, with these
values the tests were also made, as it is explained in the article; the purpose of changing these values was to corroborate the existence of the chaotic behavior in a quantitative way, through the change in the values of the exponent of Lyapunov with the decrease of the dimension of embebimiento, and qualitatively, when observing the invariant behavior in the graphs of the exponent. 3) Change in the radius of the neighborhood in which the reference point was chosen; this radius should be as small as possible but large enough so that on average each reference point has at least some neighbors. 4) Reference points, initially 500 points are an appropriate choice but should be changed if the data are intermittent or the computation time is fast and 5) Neighbors close to these points.

*20. Some assumptions are stated in various sections. Justifications should be provided on these assumptions. Evaluation on how they will affect the results should be made.*

**Reply**: Added to the text, see page 3, 5, 6, 7-10, 14-15, line 88-94, 114-116, 149-164, 178-184, 191-205, 214-216, 220-225, 238, 305-311, 320-334, 338-343

*21. The discussion section in the present form is relatively weak and should be strengthened with more details and justifications.*

**Reply**: Added to the text, see discussion section

*22. Moreover, the manuscript could be substantially improved by relying and citing more on recent literatures about contemporary real-life case studies of modelling techniques in hydrologic engineering such as the followings:*

**Reply**: The authors are grateful for the comment, the following references were taken into account

Taormina, R., et al., "Neural network river forecasting through base flow separation and binary-coded swarm optimization", Journal of Hydrology 529 (3): 1788-1797 2015.

Gholami, V., et al., "Modeling of groundwater level fluctuations using dendrochronology in alluvial aquifers", Journal of Hydrology 529 (3): 1060-1069 2015.

Chen, X.Y., et al., "A comparative study of population-based optimization algorithms for downstream river flow forecasting by a hybrid neural network model," Engineering Applications of Artificial Intelligence 46 (A): 258-268 2015.

Wang, W.C., et al., "Improved annual rainfall-runoff forecasting using PSO-SVM model based on EEMD," Journal of Hydroinformatics 15 (4): 1377-1390 2013.

Wu, C.L., et al., "A flood forecasting neural network model with genetic algorithm," International Journal of Environment and Pollution 28 (3-4): 261-273 2006.

Chau, K.W., et al., "A split-step particle swarm optimization algorithm in river stage
forecasting," Journal of Hydrology 346 (3-4): 131-135 2007.

*23.*    ***In the conclusion section, the limitations of this study, suggested improvements of***
  ***this work and future directions should be highlighted.***

**Reply**: Already added to the text, see page 16-17, line 355-361.

---

## Author Response (AR3)

1    Dear Prof. Vicente Perez-Munuzuri
2    Editor
3    Nonlinear Processes in Geophysics
4

5    You will find ci-joint the new version of our manuscript "Nonlinear analysis of the
6    occurrence of hurricanes in the Gulf of Mexico and the Caribbean Sea". As the second
7    referee proposed, we made a forecast using nonlinear analysis and maximum entropy
8    spectral decomposition from 2013 to 2017 (see page 16 and 17, lines 349 to 371).

9

10   David Salas